# Determinants of Land Use/Cover Change in the Iberian Peninsula (1990–2012) at Municipal Level

**David Fernández-Nogueira *** and **Eduardo Corbelle-Rico**

Land laboratory (LaboraTe), Department of Agricultural and Forest Engineering,
Universidade de Santiago de Compostela, 15782 Santiago de Compostela, Spain; eduardo.corbelle@usc.es
* Correspondence: david.fernandez.nogueira@usc.es; Tel.: +34-982-823-324

**Abstract:** This work analyzes the determinants associated with main land use/cover changes in the Iberian Peninsula during the 1990–2012 period using a decision tree model. Our main objective is to identify broad-scale patterns that associate the characteristics of geographic areas with the dominant land use/cover change process based on CORINE Land Cover (Coordination of Information on the Environment) and defined in a previously published work. Biophysical, structural and socioeconomic variables were considered as potentially explanatory of the dominant change process at municipal scale. The resulting model allowed identification of a common pattern in Portugal and Spain (urbanization being highly associated to areas already densely populated in the previous period), but also some diverging ones. In particular, dominant trends in Portuguese territory appear to be highly determined by wildfire occurrence. In contrast, Spanish municipalities showed more diverse patterns, usually associated to biophysical determinants like average forest productivity or average terrain slope.

**Keywords:** CORINE Land Cover; land use/cover changes; decision tree; Iberian Peninsula

---

## 1. Introduction

The European South has been one of the hotspots of land use/cover change for the last decades. The intensity and velocity of the change processes has been very significant, threatening the environmental sustainability of the different states in the Mediterranean basin [1–6]. Although the transformations have concerned most of the common land uses/covers, those that implied changes in the intensity of use of agricultural areas have probably been among the most significant, at least by total area affected [7,8]. Added to the problems commonly associated to agricultural intensification in other European regions (e.g., nitrogen pollution), in Southern regions the risks that arise from over-exploitation of water resources, erosion and desertification need to be considered [9]. Agricultural intensification, however, has taken place along with marginalization of remote areas and abandonment of former farmland, which in these regions is usually associated with wildfire risk [10,11]. Finally, other patterns of change, although less relevant in terms of total area, also threaten conservation of valuable habitats and natural resources: particularly, the expansion of impervious (urbanized) covers around metropolitan areas and in coastal zones [12–15].

Specifically concerning the Iberian Peninsula, several authors have supported the idea of an acceleration of land use/cover changes after the accession of Spain and Portugal into the European Union (then EEC) in 1986. Both countries entered the EEC just a few years before the 1990s reform of the Common Agricultural Policy (the so-called MacSherry reforms). These reforms included a number of measures oriented to decrease the overall output of the agricultural sector: subsidies for early retirement of farmers and afforestation of former agricultural land (Council Regulation 2080/1992), and lower support for prizes of agricultural products. All these measures contributed to accelerate the

already ongoing trend of farm closing and marginalization of the agricultural sector in less favored areas [16–18].

Since that moment, important transformations of forest areas have taken place, although in opposite directions in both countries. On the one hand, deforestation processes seem to have dominated Portuguese territory, which has caused a slight decline in the forest area of the country [19,20]. On the other hand, in Spain the combination of man-made afforestation and spontaneous vegetation encroachment has favored the increase of wooded area [21]. The trends followed by agricultural area were similar in both countries (net loss of total area) but this conceals, nevertheless, two opposing trends: agricultural extensification, and eventual abandonment, has affected many inland, remote or mountainous regions, like the ones dominated by dehesa/montado systems [22,23]. In contrast, an expansion and intensification of agricultural activity has taken place in the main river basins and coastal municipalities, often associated with greenhouse crops [24,25]. Finally, urban expansion has been rather important in both countries, showing urbanization rates higher than most other European countries as a consequence of relatively late (within the European context) internal migration from rural to urban areas, and to the expansion of mass tourism and real estate speculation [15,26,27].

The main objective of this study is to explore the spatial determinants associated with dominant processes of land use/cover change observed in the Iberian Peninsula at local (municipality) level. The underlying research question is whether it is possible to identify large patterns that associate the characteristics of each geographic area with a given type of dominant land use/cover change process. To carry out this work we relied on an already published analysis of the main trends of change at the municipal level in Spain and Portugal during the period 1990–2012 [24]. Starting from the classification of the municipalities presented in the aforementioned work, the objective of this work was to identify biophysical and socioeconomic variables which allowed us to better understand why municipalities followed one path or another.

## 2. Materials and Methods

The starting point for this work is the classification of the Local Administrative Units at LAU2 level (municipalities in Spain, parishes in Portugal) of the whole of the Iberian Peninsula (12,099 total units). Each local administrative unit was assigned to one of seven possible categories that differ from each other by the dominant process of land use/cover change in that period: (1) Afforestation (net gain in forest area produced by the change of agricultural-shrubland towards wooded covers) and changes in forest composition; (2) expansion of impervious surfaces; (3) stability; (4) deforestation (net loss of wooded area); (5) agricultural extensification; (6) mixed trends (afforestation, conversion to agricultural areas and farmland abandonment); and (7) agricultural intensification. The methodology and main results of this classification have already been published by Fernández-Nogueira and Corbelle-Rico [24]

The aforementioned classification was treated as dependent variable for the fitting of a classification tree. Classification/decision trees are appropriate explanatory models when the dependent variable is categorical, as in this case, while the independent variables can be categorical or numerical. They are frequently used in the literature to explore land use/cover changes e.g., [3,28–32]. Although other alternative models (e.g., random forest models or neural network models) usually result in higher classification accuracy overall, their inherent complexity hinders interpretation of the results, which goes against the objectives of this work (the generation of explanatory hypotheses of relationships between explanatory and dependent variables). Among the different methods available, we decided to use the J48 partition algorithm available in the Rweka package [33,34] for the statistical analysis language R [35].

A set of explanatory variables was prepared, including variables directly taken from public sources of statistical or cartographic data for both countries, or calculated from them. Whenever possible, the level of spatial detail used was LAU2 (municipalities in Spain, parishes in Portugal). When the original information is presented at more aggregated levels, the immediately superior level available was used (Nomenclature of Territorial Units for Statistics NUTS3 provinces, in Spain;

LAU1, municipalities, in Portugal). For the selection of potentially explanatory variables we followed the existing literature on land use/cover changes in Southern Europe and in the Iberian Peninsula, trying to include biophysical, structural, political and socioeconomic variables. The set of 17 variables used and a brief description of their original source and original resolution is shown in Table 1. "Country" was also used as a dummy variable to account for differences in the implementation of policy (e.g., common agricultural policy measures that allowed for independent decisions in member states). For the different spatial analysis necessary to compute some of the variables from the available information we used QGIS v2.18 [36] and GRASS GIS v7.6 [37].

**Table 1.** Variables introduced in the model.

| Variable | Calculated at Level | Units | Original Resolution | Source |
|---|---|---|---|---|
| **Biophysical** | | | | |
| Average elevation | LAU2 | Meters | $250 \times 250$ m$^2$ | Reuter et al. 2007 Doi:10.1080/13658810601169899 |
| Average slope | LAU2 | % | $250 \times 250$ m$^2$ | Reuter et al. 2007 Doi:10.1080/13658810601169899 |
| Average annual forest productivity 2000–2010 | LAU2 | m$^3$/ha/year | $1 \times 1$ km$^2$ | Verkerk et al. 2015 Doi:10.1016/j.foreco.2015.08.007 |
| Average climatic classification Koppen-Geiger 1981–2010 | LAU2 | Koppen table [1] | $1 \times 1$ km$^2$ | Kriticos et al. 2012 Doi:10.1111/j.2041-210X.2011.00134.x |
| Water holding capacity of soils | LAU2 | WHC table [2] | $250 \times 250$ m$^2$ | https://www.nrcs.usda.gov/wps/ portal/nrcs/detail/soils/use/ worldsoils/?cid=nrcs142p2_054022 |
| Average aridity index 1970–2000 | LAU2 | Aridity table [3] | $1 \times 1$ km$^2$ | Zomer et al. 2008 Doi:10.1016/j.agee.2008.01.014 |
| Average annual rainfall 1981–2018 | LAU2 | mm | 5 km$^2$ | Funk et al. 2014 Doi:10.3133/ds832 |
| **Structural** | | | | |
| Travel time to mayor cities >50.000 | LAU2 | Minutes | $1 \times 1$ km$^2$ | Nelson, 2008 Doi:10.2788/9583 |
| Coast/inland municipality | LAU2 | – | – | Own elaboration |
| Average distance (Euclidean) to main rivers (Strahler index > 7) | LAU2 | km | $250 \times 250$ m$^2$ | Own elaboration |
| **Political** | | | | |
| Natura 2000 | LAU2 | % of total surface | $250 \times 250$ m$^2$ | Own elaboration https: //land.copernicus.eu/local/natura |
| **Socio-economic** | | | | |
| Population density 1991 | LAU2 | Hab/km$^2$ | | National Institute of Statistics (ES) https://www.ine.es//Statistics Portugal (PT) https://portal-rpe01.ine.pt/ |
| Average farm cultivated area in 1999 | LAU2 | Agricultural surface/number of exploitations | | Agricultural Census National Institute of Statistics (ES) https://www.ine.es//Statistics Portugal (PT) https://portal-rpe01.ine.pt/ |
| Aging index 1990 | NUTS3 | % | | National Institute of Statistics (ES) https://www.ine.es//Statistics Portugal (PT) https://portal-rpe01.ine.pt/ |
| Livestock units 1990 | LAU1 (PT)/ NUTS3 (ES) | Heads/ agricultural surface | | Agricultural Census National Institute of Statistics (ES)/ Statistics Portugal (PT) https://portal-rpe01.ine.pt/ |
| Burnt areas 1990–2012 | LAU1 (PT)/ NUTS3 (ES) | Annual % | | Instituto Conservação da Natureza e das Florestas (PT) https://www.icnf.pt//Ministry of Agriculture, Fisheries and Food (ES) https://www.mapa.gob.es/es/ |
| Unemployment rate 2001 | LAU1 (PT)/ NUTS3 (ES) | % | | National Institute of Statistics (ES) https://www.ine.es//Statistics Portugal (PT) https://portal-rpe01.ine.pt/ |

[1] Koppen: C (Temperate)/B (Dry)/D (Continental with cold winters). [2] Water Holding Capacity: Ice/Glacier/ Ocean/Inland Water Bodies—Low (<25 mm)/Moderate (25–100 mm)/High (100–200 mm). [3] Aridity: (<0.03) Hyper Arid/(0.03–0.2) Arid/(0.2–0.5) Semi-Arid/(0.5–0.65) Dry sub-humid/(>0.65) Humid.

As it happens with other statistical models, one of the usual problems of recursive partition algorithms used for the generation of classification/decision trees is that of overfitting the model to the sample observations. When overfitting takes place, the resulting model often shows a considerable accuracy on the training sample, as it is able to incorporate many of the particularities of the sample. A number of pruning methods (i.e., methods to reduce the number of branches of the resulting tree and, therefore, its overall complexity) are available that can be used to prevent overfitting. In the case of algorithm J48 it is possible to modify the minimum number of observations ($M$) that each terminal node of the tree must contain: higher values of $M$ reduce the complexity of the resulting tree. We randomly divided the overall sample into training (50%) and validation (50%) subsamples to explore the appropriate value of parameter $M$. Figure 1 shows the accuracy (Cohen's Kappa) of training and validation subsamples for different values of $M$. The figure shows evidence of overfitting (training accuracy much greater than validation accuracy) for values of $M$ below 100. Taking into account this information we used a conservative value of $M = 400$ for the fitting of the final model.

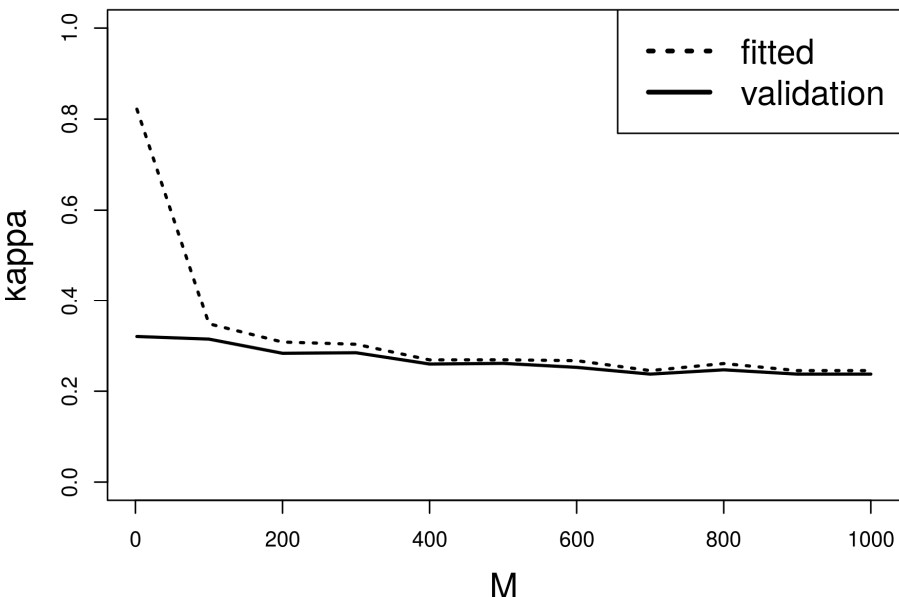

**Figure 1.** Accuracy of fitted and validation (Cohen's Kappa) depending on $M$ parameter using 50% of observations as validation sample.

It is worth mentioning that the interpretation of accuracy just presented is associated to the use of the J48 model as a hard classifier: according to this, each municipality or parish is assigned a single class, depending on the values of the input variables, and may be correctly or incorrectly classified. Nevertheless, this is not the only way to interpret the results of this kind of models. Particularly, it is also possible to interpret the result as a distribution of probability of change processes (i.e., interpreting the probability that a municipality or parish belonging to a specific class), depending on the values of the input variables. In the presentation of results that follow, both approaches are used sequentially.

## 3. Results

The fitted decision tree model has a total of 27 nodes, of which 14 are final nodes (Figure 2). It correctly assigns the dominant land use/cover change process to 43% of the total sample (5203 municipalities/parishes). An analysis from the perspective of "hard" classifier allows generating the confusion matrix shown in Table 2, in which the original class of each municipality/parish is compared with the one assigned by the model, together with the associated accuracy values.

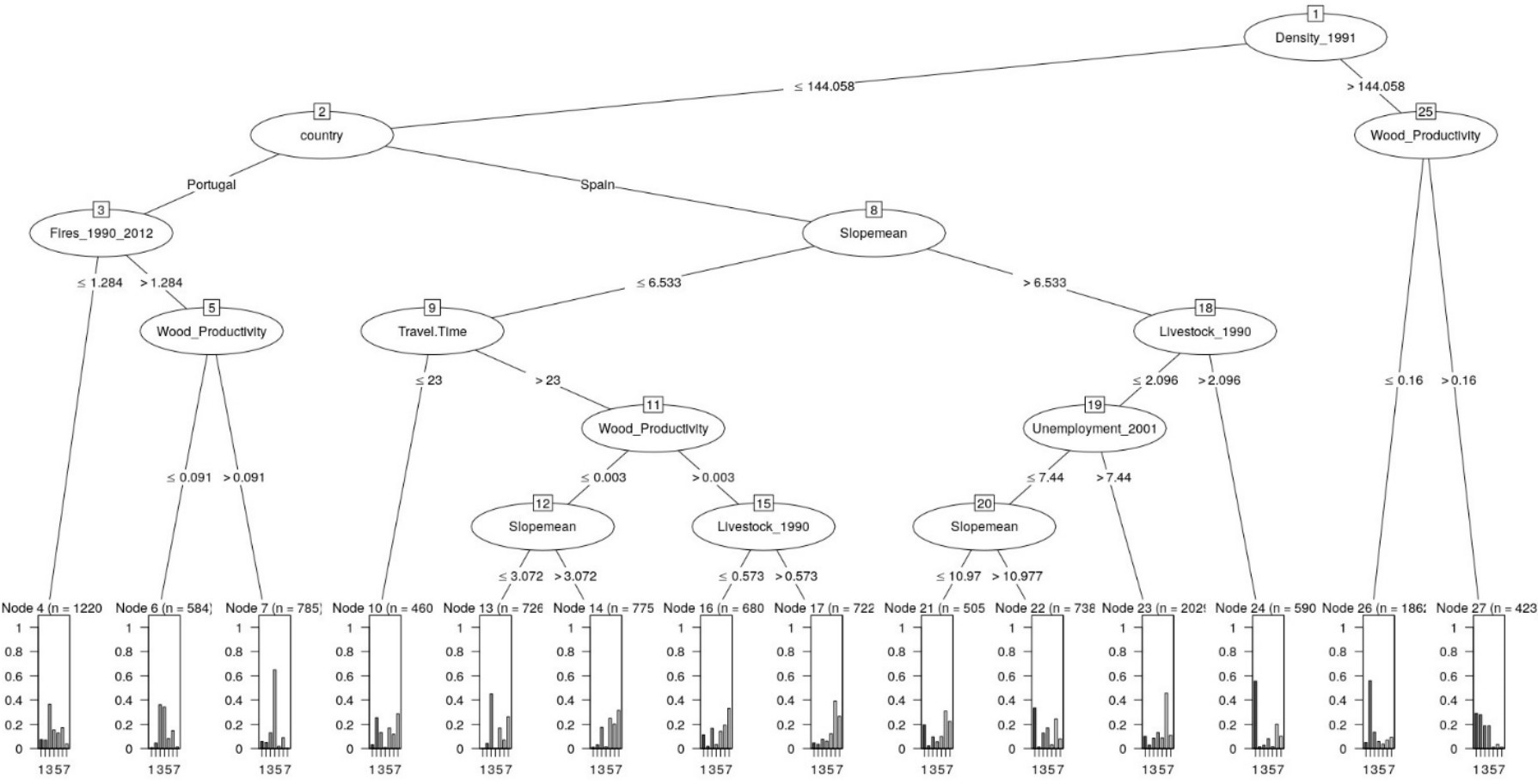

**Figure 2.** Resulting decision tree. The bar diagrams at the bottom represent the distribution of administrative units depending on dominant land use/cover process.

**Table 2.** Confusion matrix model J48.

| Original Class | Model Classification | | | | | | | Accuracy (%) | Omission Error (%) |
|---|---|---|---|---|---|---|---|---|---|
| | **1** | **2** | **3** | **4** | **5** | **6** | **7** | | |
| 1. Afforestation | 698 | 94 | 96 | 45 | 0 | 337 | 100 | 50.9 | 49.1 |
| 2. Urbanization | 136 | 1043 | 143 | 40 | 0 | 93 | 153 | 64.9 | 35.1 |
| 3. Stability | 189 | 252 | 984 | 102 | 0 | 278 | 310 | 46.5 | 53.5 |
| 4. Deforestation | 253 | 106 | 388 | 510 | 0 | 342 | 38 | 31.2 | 68.9 |
| 5. Extensification | 37 | 68 | 327 | 15 | 0 | 315 | 369 | 0 | 100 |
| 6. Mixed processes | 313 | 129 | 348 | 70 | 0 | 1365 | 343 | 53.2 | 46.9 |
| 7. Intensification | 124 | 170 | 245 | 3 | 0 | 525 | 603 | 36.1 | 63.9 |
| **Accuracy (%)** | 39.9 | 56.0 | 38.9 | 65.0 | 0 | 41.9 | 31.5 | | |
| **Commission Error (%)** | 60.1 | 44.0 | 61.1 | 35.0 | 100 | 58.1 | 68.5 | | |

The confusion matrix in Table 2 allows us to know the accuracy obtained for each of the dominant processes of land use/cover change. Thus, taking into account the errors of omission (in the case of municipalities that belong to a group in the original classification and have not been assigned), the model classifies better the municipalities dominated by urbanization processes (group 2) and afforestation (group 1) as well as the group where different processes are mixed (group 6). The behavior is similar if we address the commission error (municipalities assigned to a group that do not really belong to it): The best identified municipalities are those of groups 2 and 6, as well as group 4 (deforestation). The model seems unable to correctly identify the municipalities of group 5 (extensification of agricultural activity). A chi-squared test confirmed, nevertheless, that the classification result is different from a random model (for a confidence level lower than 0.01).

On a more descriptive note, the fitted model (Figure 2) begins with a division of municipalities according to population density in 1991: Those with higher density values would be mainly municipalities affected by urbanization (group 2). These are municipalities/parishes that coincide with the metropolitan areas (Porto-Aveiro, Lisbon, Leiria, Barcelona, Madrid, Seville and Valencia), although locations in Northern Spain (Gijón, Santander, Bilbao or Zaragoza), on the Mediterranean coast (Levante, Murcia and Andalusia) and Algarve in Portugal are also included. In other cases, a mixture appears between urbanization and afforestation processes (group 1); this seems to be the case of some municipalities located in Galicia-Basque Country, in which forest productivity was also higher.

For the remaining municipalities, the model proposes different trajectories for those located in Portugal and Spain. For Portugal, the model basically classifies municipalities into three different trajectories: Those in which wildfires were less active along the studied period (node 4) are more likely those that remained stable (group 3), mostly including areas in the Alentejo region (Setúbal, Évora, Portalegre and Beja) as well as inland areas in the Algarve. Among those municipalities in which wildfires were more active, those with higher forest productivity (node 7) are mostly those in which deforestation took place (group 4). This includes areas in all the regions of the country, to a greater or lesser extent: areas in the northern districts (Viana do Castelo, Braga and Vila Real), central areas (Viseu, Coimbra, Castelo Branco, Portalegre, Leiria and Santarém) and the southern coastline in Faro. In a mid-situation, parishes where wildfires were active but with lower forest productivity (node 6) seem to be split between deforestation and stability.

In the Spanish case, the model proposes a first division of municipalities based on the average terrain slope. With low slope values, intensification of agricultural activities (group 7) appears in all the subsequent nodes. This includes municipalities on the banks of the Guadiana (Castilla-La Mancha) and the Duero basins (León-Valladolid). It also reliably identifies the intensified areas (node 14) located in the Ebro basin and its tributaries (Navarra, La Rioja, Burgos, Huesca, Zaragoza and Lleida). The only areas of agricultural intensification that the model has not captured are municipalities associated with greenhouse intensive crops (Murcia, Almería, Cádiz or Málaga). However, although intensification is a common trend in this large group, some other trends also appear. For example, municipalities located close to large cities (node 10) are clearly also associated to urbanization processes (group 2).

This is the case of areas around Madrid, Valencia, Seville or Zaragoza and some province capitals, both inland and on the coast (e.g., Huelva, Palencia, Burgos, Valladolid, Ciudad Real, Jaén or Granada). Areas of stability (group 3) are identified in municipalities located at a greater distance from the urban centers and with low forest productivity (node 13), or located in the most important river basins in North Plateau (all of Castilla-León provinces, except Soria), inland areas of Zaragoza-Huesca and the Ebro Delta in Tarragona. However, the model fails to capture the stability of southern areas within the same node; for example, in Castilla-La Mancha and other municipalities dispersed in interior provinces. In this part of the tree, municipalities with extensification (group 5) also appear when forest productivity is low but the slope is intermediate. These are represented in various regions (nodes 10 and 14) in the peninsular center-south areas (Badajoz, Toledo, Ciudad Real, Albacete, Seville, Jaén, Granada or Córdoba), although it is also perceived in other areas of the Rioja and the Ebro Depression (Lleida, Huesca, Zaragoza and Teruel). This phenomenon is also visible (nodes 13 and 16) inland areas of Cantabria, Murcia and Castilla-León. Finally, municipalities with high average slope and high forest productivity (node 17) are often those associated to a mixture of land use/cover changes (group 6). This includes areas close to the Spanish–Portuguese border (Cáceres, Badajoz, Zamora, Salamanca) and Segovia (Figure 3).

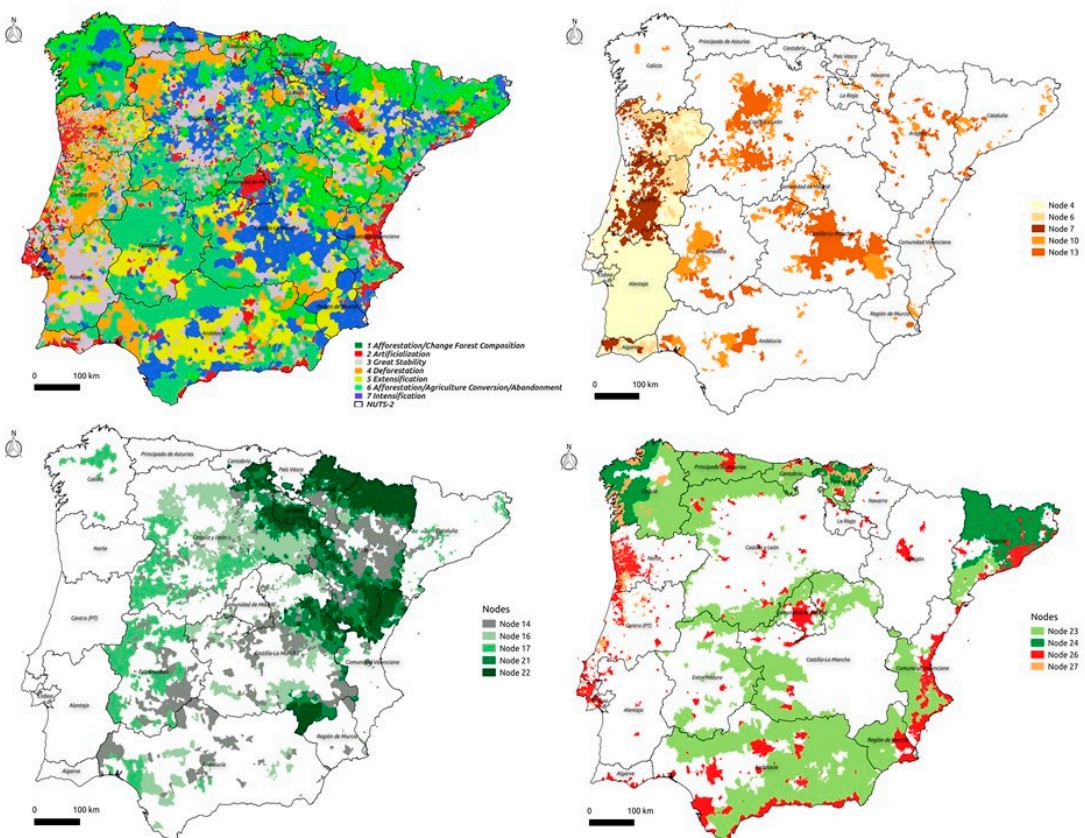

**Figure 3.** Comparison of the original classification (above, left) and the resulting nodes of the fitted tree (these are divided in three maps to improve readability).

The second great split that the model creates for Spanish municipalities is that of areas with higher average slope. Here, municipalities where afforestation was the dominant change process (group 1) form the largest group (nodes 21 and 22), along with those where the observed trend included a mixture of different change processes (group 6, node 21). The former includes areas in the Pyrenees, Cantabrian Mountains. In some areas, like the Catalonian Pre-Pyrenees (Lleida, Girona and Barcelona), Basque Country and some parts of Galicia, afforestation takes place even in presence of high livestock density (node 24).

## 4. Discussion

The model presented in this work has some problems to accurately classify a large number of LAU2 units in the Iberian Peninsula, as it can be deduced from the relatively high omission errors shown in Table 2 (Cohen's Kappa value around 0.4). Nevertheless, its performance is clearly better than what could be expected from a random model, as confirmed by the chi-squared test. Some possible explanations for this modest performance include the fact that the classification used as a starting point is itself a generalization of the dominant land use/cover change process in each administrative unit. Besides, it is evident from the results that the set of explanatory variables used is not enough to explain all the variability that is present within the study area. Nevertheless, we still believe that the results are useful, as they allow explanation of a good part of the diversity of land use/cover change dynamics in the Iberian Peninsula during the rather long period of 1990–2012 using a relatively small number of variables: population density, average terrain slope, area affected by forest fires and distance to urban centers. Most insight from the results of the fitted model can be obtained not by looking at the majority class within each terminal node of the tree, but looking at the probability distribution of categories within each terminal node. By using the results in that way, the tree allows identification of the combination of probable change processes that is associated to specific thresholds of explanatory variables.

Overall, the variables with higher influence on the type of dominant land use/change process were population density at the beginning of the period, area affected by wildfires, average slope, average woodland productivity and travel time to the nearest city. Another useful outcome is the fact that some of the potentially explanatory variables introduced in the model were not selected by the fitting process (e.g., Koppen's climate classification, water holding capacity, aridity, rainfall, river distance, Natura 2000 or aging index), which may indicate that they are less useful to explain changes at this scale.

The model offers a simple interpretation of the main trends in Portuguese territory, arguably because of its greater homogeneity, compared with the Spanish part of the Iberian Peninsula. In essence, three main trends dominated continental Portugal: stability in the Southern areas, deforestation associated to wildfires in the Northern half, and expansion of urban areas around Porto and Lisbon. On the other hand, not only was the resulting model more complex for the Spanish territory, even with its increased complexity, it had many difficulties to adequately capture its heterogeneity. Obviously, at least part of these difficulties could be attributed to the lack of additional relevant variables, or to the insufficient spatial resolution of some of the variables actually used. Examples of this limitation include the lack of municipal data from the 1989 Agricultural Census or the area affected by wildfires.

The main findings of this work are in line with the literature on land use/cover changes in the Iberian Peninsula. Namely, the increase of biomass in mountainous regions, agricultural intensification in lowlands and expansion of urbanized areas around the main cities in Spain [18] and the overall deforestation and expansion of urban areas in Portugal [19]. Other authors have already studied in detail the expansion of impervious surfaces in the metropolitan areas of both countries [16,26,38], the Mediterranean coast and the main province capitals in Spain [39].

The fact that wildfires affected large areas of Portugal during the period of study is well known, and its effect on the overall reduction of forest area has already been pointed out [20]. Forest plantations, particularly of fast-growing species like maritime pine (*Pinus pinaster*) and eucalyptus (mainly *Eucalyptus globulus*) [40] and spontaneous encroachment (more frequent in the northeastern mountainous areas of the country) [41] have not been enough to counterbalance the effect of wildfires, although this is debated by some sources [16]. On the contrary, afforestation is one of the main processes in Spain, particularly in the more rugged, mountainous areas located at higher altitude [42], but also in areas where forest plantations coexist with high livestock density (Galicia, Basque Country, Catalonia). In the latter, a significant amount of fodder (e.g., soy and corn) is imported each year, thus freeing land that should be otherwise dedicated to fodder production for other uses, essentially forest plantation of fast-growing species [43]. This is not to say that deforestation is absent from Spanish territory, very often associated to wildfires, as in Portugal [44]. The areas that the model associates to deforestation in Spain

are concentrated in the northern half of the peninsula, in areas of the Pyrenees and Iberian System [45] and in the northwestern mountains where incendiary activity is recurrent [46].

Agricultural areas have been marked by the intensification–extensification duality, but also by some instances of stability (e.g., the southern part of Portugal) [17,47] even though some traces of agricultural abandonment can be also present. In any case, the model associates agricultural intensification in Spain with flat areas in the main river basins, often associated with irrigation infrastructure [48]. The model fails to identify correctly, on the other hand, agricultural intensification associated with greenhouse crops in coastal areas along the Mediterranean [49].

One aspect that catches our attention is the accuracy with which the model identifies the trend of mixed processes. This group covers much of the territory of Spain and, to a lesser extent, Portugal. The tension between farmland abandonment, afforestation and agricultural expansion often takes place in areas where traditional uses (mostly extensive grazing) have had some weight over time, in central-western provinces [22,23] or in the main mountain systems [50].

Among the limitations of the approach used in this paper it is necessary to mention the fact that the original classification of dominant land use/cover change trends is relatively coarse (LAU2). This can explain, for example, the abundance of municipalities/parishes classified with a mixture of trends. Had the original classification made at a finer scale, we can only hypothesize how much the fit of the model would improve. Besides, it must be considered that the source on which the classification was built CORINE Land Cover (Coordination of Information on the Environment) is obviously not totally exempt of errors and methodological changes [12,51–53], which could have increased the level of noise in the classification to some extent.

## 5. Conclusions

This paper attempts to identify the spatial determinants related to the main processes of land use/cover change Iberian Peninsula during the period 1990–2012. The adjusted model suggests that a relatively limited set of spatial variables allow explanation, to a reasonable extent, of the location of the main trends of change. The model identifies similarities and differences between the territories of Spain and Portugal: in both countries, the most densely populated areas in 1991 correspond essentially to those that experienced an increase in the urbanized area in the subsequent period. Regarding differences, the dominant trends in Portugal seem to be largely conditioned by wildfires during the period of study that affected, for the most part, the northern and central part of the country. However, in Spain the model shows a stronger dependence of biophysical variables such as the average terrain slope or the average forest productivity. Areas with steeper slopes often appear dominated by afforestation, sometimes by a mixture of agricultural abandonment, afforestation and agricultural expansion. The model also recognizes the fact that afforestation can coexist with high livestock densities. On the other hand, municipalities in areas of smoother slopes tend to be dominated by the intensification of agricultural activities.

**Author Contributions:** Conceptualization, D.F.-N. and E.C.-R.; Data curation, D.F.-N.; Formal analysis, D.F.-N.; Validation, E.C.-R.; Visualization, D.F.-N. and E.C.-R.; Writing—original draft, D.F.-N.; Writing—review & editing, E.C.-R. All authors have read and agreed to the published version of the manuscript.

**Funding:** This research received no external funding.

**Conflicts of Interest:** The authors declare no conflict of interest.

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
