# Peer review of "Determinants of Land Use/Cover Change in the Iberian Peninsula (1990–2012) at Municipal Level"

_land, doi:10.3390/land9010005_

Round 1
Reviewer 1 Report
The paper “Determinants of land use/cover change in the Iberian Peninsula (1990-2012) at municipal level” analyzes LULC in Iberian Peninsula in past twenty years.
The introduction is a little bit confused a lot of situations concerning land use/cover change has been described, in a vague way. For instance authors stated that after 1986 with the entry of Spain into the European Union land use/cover changes increased without explaining the reasons (funding in several sectors, general growth of economy, etc. ). It could be better to describe trends including the values taken form reports.
Also the description of methodology can be improved. To this aim a flowchart able to synthetize the methodology can be useful.
More graphs describing trends can be useful in understanding the research.
Author Response
Reviewer 1
The paper “Determinants of land use/cover change in the Iberian Peninsula (1990-2012) at municipal level” analyzes LULC in Iberian Peninsula in past twenty years.
We want to give our most sincere thanks to the reviewer for the time and dedication provided to this manuscript.
The introduction is a little bit confused a lot of situations concerning land use/cover change has been described, in a vague way. For instance authors stated that after 1986 with the entry of Spain into the European Union land use/cover changes increased without explaining the reasons (funding in several sectors, general growth of economy, etc. ). It could be better to describe trends including the values taken form reports.
Thanks for calling our attention to this issue. Our aim when writing the introduction was to state that land use changes are a complex issue with many different trends (intensification and extensification) acting at the same time, even in close locations, and that changes affecting agricultural and forest areas are usually more relevant because of the total area affected in each case. We agree that the reference to the changes after the accession to EEC deserved a more in depth explanation and we have rewritten the text to improve it in the direction mentioned by the reviewer.
Also the description of methodology can be improved. To this aim a flowchart able to synthetize the methodology can be useful.
Thanks for this suggestion. We made some small changes in the text with the aim to improve its readability. Nevertheless, we believe that the methodology is rather straightforward: basically, a classification of LAU2 units was already available, so we gathered a number of explanatory variables and used them to fit a classification tree. We feel that a flowchart would be too linear and therefore would probably not greatly contribute to the readers’ understanding of the text.
More graphs describing trends can be useful in understanding the research.
Thank you. We acknowledge that more graphic representations can contribute to the understanding of the results. We included a new figure (fig. 3) showing the comparison between the original classification and the different areas identified by nodes of the tree model. We are open to new suggestions regarding additional figures or tables.

Reviewer 2 Report
Please, improve the English in lines: 33, 44, 51, 62 (study instead of document), 66 (land use/cover), 67-70-71 (use past tense), 77. L. 108: add software version L. 119: Why 70%-30% instead of 50%-50%? Please, explain. Table 1: Please, add URLs with source data to each row. L. 131-132: Do you mean correctly or incorrectly classified? Table 2: What happened to class 5? is it never assigned in any classification? Why could be that? Except for class 2, omission errors are very high for all other classes. A random model could be equally certain. Please, explain more in detail what is your classification useful for. Did you perform bootstrapping? How else could you consider your results as fuzzy? Did you take into account spatial autocorrelation in your model? If not, why not? I think spatial autocorrelation could explain much of the results. Did you try a machine learning technique, such as randomforest? If not, why not? L. 156: delete "correctly" L. 187-189: How do you explain this misclassified results? Why do you include country as an explanatory variable? This could be adding noise and bias to the model. I suggest to make separate models by country or better one single model regardless of the country, since this variable by itself lacks of explanatory power. If you keep it as it is, explain your reasons in the MS. Discussion: Please, add a paragraph explaining the importance of the different variables used. You used several variables initially that are not mentioned later on. L. 230-233: Where are the soft classifier results? which soft classifier did you use? L. 263: add a citation supporting your statement. The "mixed processes" class nr. 6 is not very informative and could be probably reduced by using smaller sampling units with a dominant change trend. Why did you select LAU 2 units?Author Response
Reviewer 2
Please, improve the English in lines: 33, 44, 51, 62 (study instead of document), 66 (land use/cover), 67-70-71 (use past tense), 77.
We want to give our most sincere thanks to the reviewer for the time and dedication provided to this manuscript. We fully agree with the reviewer and we made the suggested changes in those lines.
L. 108: add software version
Thank you. We included the versions of QGIS v2.18 and GRASS v7.6 in the text.
L. 119: Why 70%-30% instead of 50%-50%? Please, explain.
Thank you very much for this suggestion. Being a rather large sample (ca. 12000 observations), a validation subsample of 30% (around 4000 observations) seemed to us to be large enough in order to detect overfitting. In any case, we agree with the reviewer that a 50%-50% is more usual and therefore we did the calculations again following this scheme. The results (quite similar to the ones obtained with the 70-30 scheme) are now included in the text and in Fig. 1.
Table 1: Please, add URLs with source data to each row.
Thank you very much for these constructive comment. The DOI and URL links have been added to table 2 in each case.
L. 131-132: Do you mean correctly or incorrectly classified?
Thank you. We have changed those lines.
Table 2: What happened to class 5? is it never assigned in any classification? Why could be that? Except for class 2, omission errors are very high for all other classes. A random model could be equally certain. Please, explain more in detail what is your classification useful for. Did you perform bootstrapping? How else could you consider your results as fuzzy? Did you take into account spatial autocorrelation in your model? If not, why not? I think spatial autocorrelation could explain much of the results. Did you try a machine learning technique, such as randomforest? If not, why not?
Thank you for so many inspiring suggestions. We acknowledge that the model performance is rather poor in quantitative terms (we included a explicit mention to this issue in the Discussion section). The model, nevertheless, is clearly better than a random model and a explicit mention to a Chi-squared test was included in the Results section to support this statement.
Although its quantitative performance is less than impressive, we believe that the model is particularly useful for grabbing a general picture of the land use/cover changes in the whole peninsula, as it allows to identify associations between some explanatory variables and a number of dominant change processses. Instead of just looking at the majority class in each final tree node, we paid attention to the distribution of probability of different change processes in each final node. A explicit mention to this approach to interpret the model results is included in the Methods and in the Discussion sections. In the original text, we referred to this approach as a way of looking at the results as a “soft” or “fuzzy” classification. We acknowledge that both terms may not be the most appropriate for this situation, and therefore removed all references to them in the revised version of the manuscript.
Relative to the use of more sofisticated machine learning models, like random forests, a mention was already included in the original text in section 2. We opted for a simpler, less capable, model as we were more interested in the interpretation of the variables and threshold values selected than in the model’s overall accuracy.
L. 156: delete “correctly”
Thank you. This term has been removed from the text.
L. 187-189: How do you explain this misclassified results? Why do you include country as an explanatory variable? This could be adding noise and bias to the model. I suggest to make separate models by country or better one single model regardless of the country, since this variable by itself lacks of explanatory power. If you keep it as it is, explain your reasons in the MS.
Thank you. A dummy variable for country was introduced in the model to account for differences in the implementation of national and European policies. We acknowledge that this was not explicitly mentioned in the text and introduced some new text to do so. On the other hand, the result of fitting two separate models (one for each country) is actually the same: each of the new models has exactly the same nodes and thresholds as the corresponding branch in the model with the two countries.
Discussion: Please, add a paragraph explaining the importance of the different variables used. You used several variables initially that are not mentioned later on.
Thank you. A small description has been added in the discussion section about the variables selected and not selected by the model.
L. 230-233: Where are the soft classifier results? which soft classifier did you use?
We acknowledge that the use of the term “soft” of “fuzzy” is probably not the best in this context. Therefore, all mentions were eliminated from the text.
L. 263: add a citation supporting your statement. The “mixed processes” class nr. 6 is not very informative and could be probably reduced by using smaller sampling units with a dominant change trend. Why did you select LAU 2 units?
Thanks for this suggestion. We included a new citation in text to support that statement [number 43 in references]. In relation to the size of the sampling units, we agree with the reviewer that a smaller size would probably reduce confusion in the resulting classification. Nevertheless, we had both practical and conceptual reasons to used LAU 2 as sampling unit. First, the work is based on an already published classification that was carried out at this level, which allowed us to build upon previous results and reduce the workload for this study. Second, most socio-economic statistical information is produced for different administrative levels and LAU 2 is the lowest level in the European Unit.

Reviewer 3 Report
This paper is well-written, with substantial introductory and background information, clearly explained data and methods, and robustly interpreted results and discussion.
My major issue with it is that the results describe rather low performance of the classifier, and I wonder if there are ways in which the method might be altered to improve upon this information.
For example, I do not understand why information was aggregated to the administrative units, rather than applied to some spatial unit that pertains to the biophysical characteristics (e.g. 250m or 1km/pixel). I recognize the explanatory potential of working with the administrative units, but why would it not be preferable to say that a particular administrative unit has some proportion of afforestation and some proportion of extensification, rather than labeling the whole thing one or the other and contributing to the lower accuracy rates?
If this method truly is better, in spite of the low accuracies, then it should be substantially justified as being so. Otherwise, to hold these results to the widely used accuracy standards set by Anderson 1976 of .85 (per map, therefore potentially .578 for a change analysis), this suite of results is wanting in many categories.
For an otherwise well-conducted study, I encourage the authors to improve the method or strongly justify how it can be a useful heuristic when the resulting product is more often incorrect than correct.
Author Response
Reviewer 3
This paper is well-written, with substantial introductory and background information, clearly explained data and methods, and robustly interpreted results and discussion.
We want to give our most sincere thanks to the reviewer for the time and dedication provided to this manuscript.
My major issue with it is that the results describe rather low performance of the classifier, and I wonder if there are ways in which the method might be altered to improve upon this information.
We basically agree with this observation. Nevertheless, we argue in the text that we opted for a relatively simple model (instead of more complex alternatives like random forest or neuronal network models) because we were more interested in the interpretation of the variables selected by the model and the threshold values used than in the model’s predictive accuracy. Besides, along the text we propose interprete the model by looking at the distribution of probability of the categories in each terminal node. This is in line with the aim to detect the main associations between explanatory variables and dominant land use/cover change processes, and implicitly recognizes the fact that this associations are seldom univocal at the scale of the study.
For example, I do not understand why information was aggregated to the administrative units, rather than applied to some spatial unit that pertains to the biophysical characteristics (e.g. 250m or 1km/pixel). I recognize the explanatory potential of working with the administrative units, but why would it not be preferable to say that a particular administrative unit has some proportion of afforestation and some proportion of extensification, rather than labeling the whole thing one or the other and contributing to the lower accuracy rates?
Thank you for this suggestion. This issue has also been raised by another reviewer. We agree that a smaller sampling size would probably reduce confusion in the resulting classification. Nevertheless, we had both practical and conceptual reasons to used LAU 2 as sampling unit. First, the work is based on an already published classification that was carried out at this level, which allowed us to build upon previous results and reduce the workload for this study. Second, most socio-economic statistical information is produced for different administrative levels and LAU 2 is the lowest level in the European Unit.
We particularly appreciate the last suggestion (assigning a proportion of area affected by each change process). In fact, we have considered this approach in earlier phases of the study, but we finally opted for a more synthetic, general, view (provided by the dominant change process in each location). In any case, our previous experience with land use change models at LAU 2 level does not indicate that the alternative would necessarily result in better model accuracy (see, for example, DOI: 10.1016/j.landusepol.2013.10.013), something that reflects the complexity of the links between the socio-economic and biophysical systems and the land use/change system.
If this method truly is better, in spite of the low accuracies, then it should be substantially justified as being so. Otherwise, to hold these results to the widely used accuracy standards set by Anderson 1976 of .85 (per map, therefore potentially .578 for a change analysis), this suite of results is wanting in many categories.
We appreciate the suggestion, but we do not believe that the accuracy standars for classification of remotely sensed imagery are applicable in this case. Models that include socio-economic variables usually show less impressive results in terms of accuracy, reflecting both the complexity of the processes modelled and the shortcomings of the variables used (available only for some administrative levels, usually just for some specific years …). The aforementioned paper and others could probably serve as examples: DOI: 10.1016/j.landusepol.2013.10.013 ; 10.1016/j.landusepol.2012.06.011 .
For an otherwise well-conducted study, I encourage the authors to improve the method or strongly justify how it can be a useful heuristic when the resulting product is more often incorrect than correct.
Thank you. We still believe that, although not impressive in terms of final accuracy, the model can be useful for the purpose of the paper (“ identify large patterns that associate the characteristics of each geographic area with a given type of dominant land use/cover change process”). Nevertheless, we agree with the reviewer that this was not clearly expressed in the text. We included changes in the Discussion section to reflect this usefulness in a more explicit view.

Round 2
Reviewer 2 Report
Authors have addressed most of the requirements and the paper is now ready for publication.